# Availability of Third Molars as Donor Teeth for Autotransplantation to Replace Congenitally Absent Second Premolars in Children and Young Adults

**DOI:** 10.3390/diagnostics13111874

**Published:** 2023-05-27

**Authors:** Małgorzata Bilińska, Tomasz Burzykowski, Paweł Plakwicz, Małgorzata Zadurska, Ewa Monika Czochrowska

**Affiliations:** 1Private Practice, 02-916 Warsaw, Poland; m.b.bilinska@gmail.com; 2Data Science Institute, Hasselt University, 3500 Hasselt, Belgium; 3Department of Statistics and Medical Informatics, Medical University of Bialystok, 15-295 Białystok, Poland; 4Department of Periodontology, Medical University in Warsaw, 02-097 Warsaw, Poland; 5Department of Orthodontics, Medical University in Warsaw, 02-097 Warsaw, Poland; malgorzata.zadurska@wum.edu.pl

**Keywords:** agenesis, autotransplantation, hypodontia, second premolars, third molars

## Abstract

The aim of the study was to assess the presence and distribution of third molars (M3) regarding their autotransplantation in patients with congenital absence of second premolars (PM2). Additionally, M3 development in relation to patients’ age and gender was investigated. Panoramic radiographs of non-syndromic patients with at least one congenitally absent PM2 were used to assess the localization and number of missing PM2 and the presence or absence of M3 (minimum age 10 years). The alternate logistic regression model was applied to analyze associations between the presence of PM2 and M3. A total of 131 patients with PM2 agenesis were identified (82 females, 49 males). At least one M3 was present in 75.6% and all M3 were present in 42.7% of patients. A statistically significant association between the number of PM2 and M3 agenesis was found; the effects of age and gender were not significant. More than half of M3 in patients between 14–17 years old had completed ¼ of their root development. The congenital absence of maxillary PM2 was associated with the absence of maxillary PM2, M3, and no correlation was found in the mandible. In patients with PM2 agenesis, at least one M3 is often present and can be considered as a donor tooth for autotransplantation.

## 1. Introduction

Hypodontia (tooth agenesis) is a distinct pattern of the congenital absence of at least one permanent tooth or a tooth germ. The condition is considered as the most common dental anomaly in humans [1]. Mandibular second premolars (PM2) are the most frequently congenitally missing teeth, except third molars (M3) [2,3]. The meta-analysis by Polder et al. confirmed that the prevalence of PM2 agenesis occurred in more than 60% of patients with tooth agenesis, excluding M3 agenesis [4]. Indeed, 21% of the affected patients were missing maxillary PM2, in contrast to 41% who were missing mandibular PM2. Bilateral tooth agenesis mostly affects mandibular PM2, followed by either maxillary PM2 or maxillary lateral incisor agenesis [5,6].

The congenital absence of PM2 can cause tilting and migration of the neighboring teeth, and thus negatively affects the occlusion. This can be especially important in patients with mandibular retrognathia and a deep-bite tendency. Possible treatment options include the preservation of primary molars, dental implants or other prosthodontic restorations, orthodontic space closure, hemisection and tooth autotransplantation [7]. The long-term preservation of primary molars in patients with PM2 agenesis has been previously reported, however, the authors stated that it is not possible to predict the probability of survival for a single primary molar at an early age [8,9]. Caries and root resorption, especially ankylosis of primary molars, is also a serious concern for their long-term preservation [10]. Dental implants are generally contraindicated before growth has ceased, which is an important concern in growing patients with premolar agenesis and an absence of or a bad prognosis for primary second molars [11,12]. Orthodontic space closure in patients with PM2 agenesis depends on the occlusion, space conditions, profile and patients’ wishes. Space closure to treat the congenital absence of PM2 is generally more difficult in the mandible, even considering the application of skeletal anchorage (TADs) [13]. A hemisection of primary second molars can be utilized to facilitate orthodontic space closure to treat premolar agenesis, but it requires the presence of a healthy primary molar [14].

Tooth transplantation, especially the autotransplantation of developing premolars, is a valid option to treat the congenital absence of premolars [15,16]. The best results regarding pulpal and periodontal healing have been obtained for premolars with developing roots at 1/2 to 3/4 of the final root length at the time of surgery [17]. However, orthodontic indications or a possibility of premolar extraction at the recipient site should be present. Moreover, there is a limited time period related to the premolar root development, when it can serve as a donor tooth for autotransplantation. The autotransplantation of mature premolars can be also considered, but its outcome is less predictable than the transplantation of developing premolars [18]. These are important limitations for the use of autotransplantation of developing premolars to treat PM2 agenesis.

The autotransplantation of a developing M3 is an attractive alternative to replace missing PM2 because their removal does not affect the occlusion in the same manner as the removal of a premolar. M3 teeth are also often extracted due to space deficiency in posterior segments. The selection of M3 as a donor tooth for transplantation is better accepted by patients and their parents because these teeth are somehow regarded as spare teeth in an oral cavity. However, the prevalence of the congenital absence of M3 has been reported as high as 20–30% and has also been correlated with the congenital absence of other teeth [6,19,20,21,22,23]. This may present an important problem in patients with a congenital absence of PM2 because, in these patients, M3 might be absent as well. Evidence regarding the presence of M3 in patients with PM2 agenesis is scarce [4,6,24].

Maxillary M3 teeth usually have a better morphology to serve as donor teeth for autotransplantation, because they are often smaller with a better root morphology (conical shape, single roots) than mandibular M3 teeth. So far, no information on the distribution of M3 teeth presence in patients with PM2 agenesis has been reported. Furthermore, the correlation between the number of congenitally absent PM2 and M3 teeth has not been previously examined. Therefore, the aim of the study was to assess the presence and distribution of M3 teeth in patients with a congenital absence of PM2 teeth as possible donors for autotransplantation. An additional aim was to investigate the development of M3 teeth in relation to patients’ age and gender.

## 2. Materials and Methods

All available panoramic radiographs of non-syndromic patients with at least one congenitally absent PM2 were selected from the files of the Department of Orthodontics at the Warsaw Medical University, Poland, two orthodontic and one oral surgery private practices in Warsaw and the radiology center in Warsaw. The study was approved by the Bioethics Committee of the Warsaw Medical University (No: AKBE/80/17).

The exclusion criteria comprised the congenital absence of more than 6 permanent teeth, except M3, patients with syndromes and dentofacial deformities and poor-quality images. The minimum age of a patient to be included was set at 10 years old. The localization and number of congenitally absent PM2 and the presence or absence of M3 were registered. A tooth was diagnosed as congenitally missing when no signs of crown mineralization were seen on the panoramic radiograph and no history of extraction or loss of a particular tooth was confirmed or suspected. Panoramic radiographs were examined twice after a 1-month interval by two investigators (MB, EC) and the reproducibility in the identification of the congenital absence of a tooth and the stage of M3 development was 100%.

The development of M3 was assessed using the composed scale with 6 stages of tooth formation. This scale is based on 14 stages of molar development by Moorrees: initial cusp formation (Ci), coalescence of cusps (Cco), cusp outline complete (Coc), crown half complete (Cr1/2), crown three-quarter complete (Cr3/4), crown complete (Crc), initial root formation (Ri), initial cleft formation (Cli), root length quarter (R1/4), root length half (R1/2), root length three-quarters (R3/4), root length complete (Rc), apex half closed (A1/2) and apical closure complete (Ac) [25]. The composed scale of M3 development included the following stages: crown development (Ci, Cco, Coc, Cr1/2, Cr3/4), final crown and initial root development (Crc, Ri, Cli), root length quarter (R1/4), root length half (R1/2), root length three-quarters (R3/4), final root development and apex closure (Rc, A1/2, Ac). The composed scale aimed at an evaluation of M3 development from the perspective of a possible donor tooth for autotransplantation.

### Statistical Methods

Patients’ demographic and clinical characteristics were summarized using descriptive statistics (mean, standard deviation and range for continuous characteristics; percentages for the categorical ones).

Differences in the distribution of categorical outcomes, as observed in contingency tables, were compared by applying the chi^2^ test evaluated by using the 5% two-sided significance level. The logistic regression model was used to investigate the dependence of the probability of a binary outcome (as the presence of all M3) on covariates. Tests of the statistical significance of the effects of the covariates were evaluated by using the 5% two-sided significance level.

The probability of the co-occurrence of PM2 and M3 was analyzed by using the alternate logistic regression model [26]. In particular, the association between the presence of a particular PM2 and M3 was modeled by using pairwise odds ratios. Symmetric associations were assumed: for instance, the strength of the association between the mandibular right PM2 and the mandibular right M3 was assumed to be the same as the strength of the association between the mandibular left PM2 and the mandibular left M3; or, the strength of the association between the mandibular right PM2 and the maxillary left M3 was assumed to be the same as the strength of the association between the mandibular left PM2 and the maxillary right M3. Significance tests for the odds ratios were evaluated by using the 5% two-sided significance level, with the Bonferroni correction for multiple testing. All analyses were conducted by using the SAS v.9.4 software.

## 3. Results

A total of 131 Caucasian patients with a congenital absence of at least one PM2 were identified, including 82 females (62.6%) and 49 males (37.4%). The mean age was 13.94 years (SD: 2.73 years; range: 10 to 22 years). The distribution of patients according to age and gender is shown in Table 1. A total of 73 patients (56%) were younger than 14 years of age; 40% of panoramic radiographs (53 patients) were selected from the orthodontic files and 60% of radiographs (78 patients) were selected from the dental files.

### 3.1. Distribution of the Congenital Absence of PM2

Table 2 presents the distribution of congenitally absent PM2 according to the tooth number and the patients’ gender. Mandibular PM2 were more than three times more often missing than maxillary PM2.

Table 3 shows the distribution of congenitally absent PM2 according to the tooth number and gender. The majority of patients (80%) were missing one or two premolars, also within male and female subgroups. The most common pattern was a bilateral absence of mandibular PM2 followed by a unilateral absence of the right or the left mandibular PM2. The absence of all premolars was registered three times more often in females than in males (twelve versus four patients, respectively).

### 3.2. Distribution of the Presence of M3

Table 4 presents the distribution of present M3 in patients with a congenital absence of at least one PM2 according to the tooth number and gender. Maxillary M3 were more often present than mandibular M3.

Table 5 shows the distribution of present M3 in patients with a different number of congenitally absent PM2. At least one M3 was present in 75.6% of the patients and all M3 were present in 42.7% of the patients. Only one M3 was present in 10.7% of patients, two in 10.7% and three in 11.4% of patients.

Table 6 presents the distribution of patients with a different number of congenitally absent PM2 and M3. There was a statistically significant association between the number of missing PM2 and M3 (*p* = 0.009). In particular, the proportion of patients with all M3 increased with the increasing number of PM2. To investigate this association in more detail, the probability of the presence of all M3 was analyzed by using a logistic regression model (fitted using Firth’s penalized maximum likelihood) with gender, age (below or above 14 years of age) and the number of PM2 as factors. The effects of age and gender were not statistically significant (*p* = 0.222 and 0.999, respectively). The effect of the number of PM2 was statistically significant (*p* = 0.017). In particular, the odds of the presence of all M3 increased 8.2, 23.5 and 42.1 times for one, two and three PM2, respectively, as compared to the case of all absent PM2.

### 3.3. Development of M3 Related to Age

Table 7 presents the distribution of different stages of M3 formation in relation to age and gender.

The development of M3 in patients younger than 14 years did not generally exceed the stages of crown formation. In these patients, only 11.4% of M3 had root development completed at ¼–½ of the final root length, which can be estimated as minimal for considering transplantation. More than 50% of M3 in patients with PM2 agenesis between 14 and 17 years had root development completed in ¼ to ¾ of the final root length, which is presumably optimal for their successful transplantation. All patients with a congenital absence of PM2 who were older than 18 years had completed formations of M3 crowns and root formation was completed in 27.2% of them.

### 3.4. Association between Congenital Absence of PM2 and M3

Figure 1 shows the estimates of odds ratios (statistically significant at the 5% significance level, after correction for multiple testing), which describe the strength of the association between the congenital absence of PM2 and M3 based on the alternate logistic regression model.

There was a strong association between the congenital absence of M3. For instance, the odds of M3 absence in the maxilla increased 45.5 times when the other maxillary M3 was absent (equivalently, the odds of M3 presence in the maxilla increased 45.5 times when the other M3 in the maxilla was present). In the mandible, the odds ratio was equal to 75.2. Furthermore, the odds of M3 absence in the mandible increased 16.6 times when M3 on the same side was absent in the maxilla and 14.7 times when M3 was absent on the other side of the maxilla.

The odds of the congenital absence of PM2 in the maxilla increased 26.4 times when the other PM2 in the maxilla was absent (equivalently, the odds of the presence of PM2 in the maxilla increased 26.4 times when the other PM2 in the maxilla was present). In the mandible, the odds ratio for PM2 absence was not statistically significantly different from 1. In other words, it could not be precluded that mandibular PM2 are independently absent of each other. Similarly, the absence of PM2 in the mandible seemed to be independent of the absence of PM2 in the maxilla.

Finally, the congenital absence of PM2 in the maxilla was associated with M3 absence. For instance, the odds of M3 absence increased 5.7 times if PM2 was absent on the same side of a maxilla and 3.7 times if PM2 was absent on the other side of a maxilla. Moreover, the odds of M3 absence in a mandible increased 4.7 times if PM2 was absent on the same side of a maxilla and 4.1 times if PM2 was absent on the other side of a maxilla. Therefore, the absence (or presence) of PM2 in a maxilla seems to be associated with the absence (or presence) of M3.

The congenital absence of PM2 in a mandible was, in general, not associated with the absence of M3, except for the 2.5-fold increase in the odds of M3 absence on the other side of a mandible. This implies that PM2 absence in the mandible does not preclude the M3 absence in the mandible—especially if PM2 and/or M3 are present in the maxilla.

## 4. Discussion

The results of our study confirmed that, in the majority of patients (75%) with a congenital absence of PM2, at least one M3 was often present. Therefore, M3 are likely to be considered as possible donor teeth to replace congenitally absent PM2. The absence of M3 was associated with the increased number of congenitally absent PM2, which may restrict the availability of this treatment option in patients with multiple PM2 ageneses. The congenital absence of mandibular PM2 was more frequent and was also not associated with M3 absence, in contrast to maxillary PM2 congenital absence, which was correlated with M3 absence. More than 60% of included patients were females.

A higher prevalence of the congenital absence of mandibular PM2 and other teeth was reported in orthodontic patients [27]. Premolar agenesis is often diagnosed during a routine orthodontic examination on panoramic radiographs. Orthodontists should have a leading role in treatment planning for patients with congenital tooth absences. They evaluate skeletal and dental relations, space conditions and patient profiles in order to formulate and discuss available treatment options, also in cooperation with other dental specialists. Patients’ opinion is always considered to obtain informed consent.

The reported prevalence of maxillary PM2 congenital absence varied from 0.2 to 3.7% [21,27], and the prevalence of mandibular PM2 congenital absence varied from 0.7 to 6.9% [27,28]. The prevalence of PM2 agenesis was often calculated in samples, which excluded or did not include M3 agenesis, and varied from 6.2 to 51.8% for maxillary PM2 [29,30] and from 12.8 to 64.3% for mandibular PM2 [29,31]. The higher prevalence of mandibular PM2 congenital absence was also confirmed in the present study, but our results showed an even higher predominance of PM2 absence in the mandible, which was more than three times more frequent than in the maxilla: 71.8–77.9% vs. 20.7–21.4%, respectively. The prevalence of all M3 congenital absences ranged from 25.7 to 41.6% [22,32]. The prevalence of PM2 congenital absence was reported to be significantly higher in individuals with a bilateral absence of maxillary M3 or mandibular M3 in comparison to the controls [22,23].

Only two studies, to the best of our knowledge, have evaluated tooth agenesis in orthodontic patients with a congenital absence of PM2 [6,24]. Garib et al. studied the occurrence of dental anomalies associated with PM2 agenesis in 203 Brazilian orthodontic patients [24]. The prevalence of the congenital absence of M3 in 77 patients older than 14 years was 48.1%. This is a higher percentage than in our sample since agenesis of M3 ranged from 46.6 to 34.3% for teeth 38 and 28, respectively. Endo et al. analyzed tooth agenesis in a group of 80 Japanese orthodontic patients with a bilateral congenital absence of PM2 [6]. They reported a higher prevalence of M3 agenesis (63.6%) than in the present study and in the study by Garib et al. Furthermore, the prevalence of all M3 ageneses was higher compared to our results, which were 30% and 24.4%, respectively. These differences could be explained by different selection criteria, since the studies only included orthodontic patients, and Endo et al. also only evaluated patients with bilateral mandibular PM2 absence, including 12 patients with oligodontia. In the present study, a much higher number of individuals with a congenital absence of PM2 was included (133) compared to the other two studies. This number also allowed for more detailed comparisons of the distribution of PM2 and M3 agenesis, which had not been previously investigated. A similar pattern of gender distribution, with females representing about two-thirds of the patients with PM2 agenesis, was seen in all studies.

The prevalence of a congenital absence of M3 was statistically significantly increased with the increasing number of absent PM2 in our study (Table 6). It is interesting to note that patients without any M3 absence and patients with an absence of all M3 were the two most numerous groups among the M3 agenesis groups. Only about 20% of patients with a congenital absence of PM2 were missing more than two premolars, so this agenesis pattern was not common in our sample. We did not analyze the presence of M3 in patients with PM2 agenesis in relation to the concomitant agenesis of other teeth, which may play a role and be a prognostic factor in the prevalence of M3 agenesis. This could be studied in more detail in the future, but the collection of an appropriate sample size is not easily obtainable. Garib et al. found 33 patients (16.3%) with a congenital absence of maxillary lateral incisors in a sample of 203 orthodontic patients with PM2 agenesis, which is a significantly higher prevalence than in the general population [24]. Bhutta et al. established a positive association between the congenital absence of M3 and maxillary and mandibular lateral incisor agenesis in 270 orthodontic patients, but no such association was present for the PM2 agenesis sample [21].

The congenital absence of maxillary PM2 was associated with the absence of M3, while no such association was present for the agenesis of mandibular PM2 (Figure 1). This is important information regarding the possibility of the autotransplantation of developing M3, because orthodontic space closure in the mandible is more difficult regarding the anatomy of mandibular molars and possible unwanted consequences for occlusion and profile. It is also worth noting that the absence of mandibular PM2 was not associated with the absence of maxillary M3. Maxillary M3 have more a favorable morphology for transplantation than mandibular molars, since their roots are often conical roots and they are usually smaller than mandibular M3. It is better to select the donor tooth from the opposite arch in case of a failure of the orthodontic space closure and the mesialization of all molars, including M3 on the affected side. Space closure is usually easier in the maxilla because of the morphology of an upper first molar and the possibility of palatal TADs. Therefore, the associated absence of any M3 in patients with maxillary PM2 agenesis seems to be less important.

The assessment of M3 development in the present sample has shown that, in the majority of patients younger than 14 years, very few M3 started to develop roots, which generally makes M3 autotransplantation not possible in this age group. It seems that the best age to treat the congenital absence of PM2 with the autotransplantation of developing M3 is between 14 and 17 years of age. At this age, half of the M3 have their root development between ¼ to ¾ of the final root length, which is favorable for pulp revascularization. The gentle surgical removal of M3 is mandatory for the success of tooth transplantation, and it is also easier to gently remove unerupted teeth with shorter roots. Half of the patients with PM2 absence who were older than 18 years had root development of M3 not exceeding ½ of the final root length. The results of the study show great variety in M3 development, and therefore, a need for individualized treatment planning. Our findings agree with the chronology of dental development presented by Moorrees et al., who reported that second and third molars required the longest times for the formation of their crowns and roots [25]. The reported average time for the formation of an M3 root (Crc-Rc) was 4.5 years, which corresponds well with our results. However, dental implants in older patients are a valid alternative to replace missing teeth and this option should be discussed in detail with the patients.

### Limitations

Potential sources of bias in the present study include selection and information bias. We used a sample from individuals seeking dental or orthodontic treatment, and the true prevalence in the population may differ from the one recorded in our sample. However, we did include all available patients with a congenital absence of PM2 to evaluate the presence of M3 as donor teeth for autotransplantation. Comparisons between patients with and without PM2 agenesis could provide additional information on the presence and localization of M3. Such comparisons are possible in a large cohort sample, which is not easy to collect.

Patients with hypodontia were reported to have delayed dental age in comparison to non-hypodontia patients [30,33,34]. This delay was reported to be from a few months up to one year. Therefore, the number of congenitally absent PM2 can be overestimated in the present study. It is difficult to estimate the prevalence of late-developing PM2 in a population because this condition was only described in case reports or case series. Females represented more than 60% of the individuals in our sample, and since their dental development is generally ahead of males, this might possibly reduce the risk of a misdiagnosis of PM2 congenital absence. At the same time, the presence of M3 might be underestimated in patients below 14 years of age, which is a common age limit to confirm the congenital absence of M3. Moorrees et al. reported that the average time for the initial crown formation of M3 was 9.4 years [25]. Lee et al. also confirmed that the development of M3 in the Korean population was likely to begin at the age of 7 years in females and males [35]. However, studies by Garib et al. and Endo et al., which included patients older than 13–14 years, have shown a higher prevalence of M3 absence than in the present study [6,24]. The inclusion of patients younger than 14 years aimed to increase the sample size, acknowledging that late-developing PM2 are uncommon. The possible underestimation of M3 presence in our sample can only increase the possibility of M3 autotransplantation in older patients.

## 5. Conclusions

In the majority of patients with a congenital absence of PM2, at least one M3 was present and, therefore, it could be considered as an autotransplantation donor. Mostly, one or two PM2s were absent.The prevalence of a congenital absence of M3 was statistically significantly increased in relation to the increasing number of congenitally absent PM2.The congenital absence of maxillary PM2 was statistically significantly associated with PM2 and M3 absence, while the congenital absence of mandibular PM2 occurred independently from PM2 and M3 absence.The best age to perform autotransplantation of immature M3 in relation to their development (1/4 to ¾ of their final root length) is usually between 14 to 17 years.

## Figures and Tables

**Figure 1 diagnostics-13-01874-f001:**
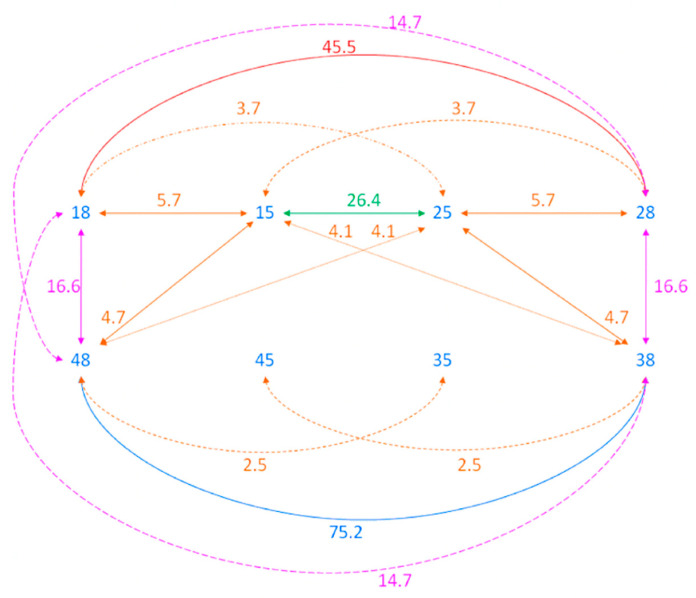
Model-based estimates of odds ratios that describe the strength of the association between the congenital absence of PM2 and M3 in the analyzed group of patients. Only the estimates that are significantly different from 1 are presented. The coloured lines show the odds ratio between the specific pairs of teeth. Example: the odds of M3 agenesis in the maxilla increased 45.5 times when the other M3 in the maxilla was missing; the odds of agenesis of M3 in the maxilla increased 16.6 times when M3 was missing on the same side in the mandible.

**Table 1 diagnostics-13-01874-t001:** Distribution of patients with a congenital absence of at least one PM2 according to age and gender.

Gender/Age	10–13 Years Old	14–17 Years Old	18–22 Years Old	Total
Females	47 (57.3%)	27 (32.9%)	8 (9.8%)	82 (100%)
Males	26 (53.1%)	18 (36.7%)	5 (10.2%)	49 (100%)
Total	73 (55.7%)	45 (34.3%)	13 (9.9%)	131 (100%)

**Table 2 diagnostics-13-01874-t002:** Distribution of the congenitally absent PM2 in relation to the tooth number and patients’ gender. The percentage of male and female patients with a particular PM2 absence was separately calculated for each gender.

Gender/Tooth	15	25	35	45
Females (*N* = 82)	22 (26.8%)	21 (25.6%)	58 (70.7%)	66 (80.5%)
Males (*N* = 49)	6 (12.2%)	6 (12.2%)	36 (73.5%)	36 (73.5%)
Total (*N* = 131)	28 (21.4%)	27 (20.7%)	94 (71.8%)	102 (77.9%)

**Table 3 diagnostics-13-01874-t003:** Distribution of congenitally absent PM2 according to the tooth number and gender. The percentage of male and female patients with different patterns of PM2 agenesis was separately calculated for each gender.

Absent Tooth	15	25	35	45	15 & 25	15 & 35	25 & 45	35 & 45	15 & 25 & 35	15 & 35 & 45	25 & 35 & 45	All Absent
Females 82/63%	1/1.2%	1/1.2%	8/9.8%	20/24.4%	2/2.4%	3/3.7%	0/0%	26/31.7%	1/1.2%	3/3.7%	5/6.1%	12/14.3%
Males 49/37%	1/2%	0/0%	12/24.5%	10/20.4%	0/0%	0/0%	2/4.1%	19/38.8%	0/0%	1/0.8%	0/0%	4/8.2%
Total 131/100%	2/1.5%	1/0.8%	20/15.3%	30/22.9%	2/1.5%	3/2.3%	2/1.5%	45/34.5%	1/0.8%	4/2%	5/3.8%	16/12.2%

**Table 4 diagnostics-13-01874-t004:** Distribution of M3 presence in patients with a congenital absence of at least one PM2 according to the tooth number and gender. The percentage of male and female patients with a particular M3 presence was separately calculated for each gender.

Gender/Tooth	18	28	38	48
Females (N = 82)	49 (59.7%)	53 (64.6%)	39 (47.5%)	42 (51.2%)
Males (N = 49)	32 (66.3%)	33 (67.3%)	31 (63.3%)	32 (66.3%)
Total (N = 131)	81 (61.8%)	86 (65.7%)	70 (53.4%)	74 (56.5%)

**Table 5 diagnostics-13-01874-t005:** Distribution of M3 presence in patients with a congenital absence of PM2 according to the tooth number and gender. The percentage of male and female patients was separately calculated for each gender.

Missing Tooth	All Absent	18	28	38	48	18 & 28	28 & 38	38 & 48	18 & 28 & 38	18 & 28 & 48	18 & 38 & 48	28 & 38 & 48	All Present
Females 82/ 63%	23/28%	2/2.4%	4/4.9%	0/ 0%	2/2.4%	9/ 11%	1/ 1.2%	1/1.2%	1/0.8%	3/3.7%	1/1.2%	2/2.4%	33/ 40.2%
Males 49/ 37%	9/ 18.4%	1/ 2%	3/6.1%	1/ 2%	1/ 2%	1/ 2%	0/ 0%	2/ 4.1%	2/4.1%	3/6.1%	2/ 4.1%	1/ 2%	23/46.9%
Total 131/100%	32/ 24.4%	3/2.3%	7/5.3%	1/0.8%	3/2.3%	10/7.6%	1/0.8%	3/2.3%	3/2.3%	6/4.6%	3/2.3%	3/2.3%	56/42.7%

**Table 6 diagnostics-13-01874-t006:** The number and percentage of patients with a different number of congenitally absent PM2 and M3.

Number of Absent PM2	Number of Absent M3					
	0	1	2	3	4	Total
1	31 (58.5%)	6 (11.3%)	3 (5.7%)	5 (9.4%)	8 (15.1%)	53 (100%)
2	23 (44.2%)	6 (11.5%)	7 (13.5%)	5 (9.6%)	11 (21.1%)	52 (100%)
3	2 (20%)	1 (10%)	2 (20%)	2 (20%)	3 (30%)	10 (100%)
4	0 (0%)	2 (12.5%)	2 ( 12.5%)	2 (12.5%)	10 (62.5%)	16 (100%)
Total	56	15	14	14	32	

**Table 7 diagnostics-13-01874-t007:** Distribution of different stages of M3 development according to age and gender.

Stage of M3 Development According to Moorreess et al [25].	10–13 years			14–17 years			18–22 years		
	All (100%)	Females (100%)	Males (100%)	All (100%)	Females (100%)	Males (100%)	All (100%)	Females (100%)	Males (100%)
Crown development (Ci, Cco, Coc, Cr1/2, Cr3/4)	74 (49.3%)	41 (44%)	33 (57.9%)	22 (18%)	8 (10.7%)	14 (29.8%)	0 (0%)	0 (0%)	0 (0%)
Final crown and initial root development (Crc, Ri, Cli)	59 (39.3%)	41 (44%)	18 (31.6%)	32 (26.2%)	23 (30.7%)	9 (19.1%)	10 (30.3%)	9 (52.9%)	1 (6.2%)
Root length quarter (R ¼)	13 (8.7%)	8 (8.6%)	5 (8.8%)	16 (13.1%)	12 (16%)	4 (8.5%)	0 (0%)	0 (0%)	0
Root length half (R ½)	4 (2.7%)	3 (0.4%)	1 (1.7%)	22 (18%)	16 (21.3%)	6 (12.8%)	7 (21.2%)	4 (23.5%)	3 (18.7%)
Root length three-quarters (R ¾)	0 (0%)	0 (0%)	0 (0%)	24 (19.7%)	16 (21.3%)	8 (17%)	7 (21.2%)	3 (17.6%)	4 (25%)
Final root development and apex closure (Rc, A½, Ac)	0 (0%)	0 (0%)	0 (0%)	6 (4.9%)	0 (0%)	6 (12.8%)	9 (27.2%)	1 (5.9%)	8 (50%)

## Data Availability

The data presented in this study are available on request from the corresponding author.

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
