# Peer review of "Availability of Third Molars as Donor Teeth for Autotransplantation to Replace Congenitally Absent Second Premolars in Children and Young Adults"

_diagnostics, 2023, doi:10.3390/diagnostics13111874_

Round 1
Reviewer 1 Report
The submitted manuscript presents detailed research about the presence of dental agenesis in children. This is also the assesment of autotransplantation possibilty in case of second premolars absence. There are minor changes necessary to make easier the text understandable.
In Results please precise definition: congenital absence of PM2. Maybe it should be congenital absence of at least one PM2. It is also applies to Table 1.
Most of tables need some editors correction. They should have rather title over the tables than caption positioned under them and always capital letter in the beginning. Also numbers of tables should be standardized: Roman or Arabic.
In Figure 1. which is very interesting and valuable , the association between hypodontia of maxillary PM2 and mandibulary PM2 is missed . If it is possible for authors it can be added.
Author Response
Dear Reviewer,
Many thanks for your time to review our paper and for your valuable comments to the manuscript.
Answers to the the reviewer's comments:
- In Results please precise definition: congenital absence of PM2. Maybe it should be congenital absence of at least one PM2. It is also applies to Table 1. Answer: We have included the suggested comments in the review of the paper.
- Most of tables need some editors correction. They should have rather title over the tables than caption positioned under them and always capital letter in the beginning. Also numbers of tables should be standardized: Roman or Arabic. 2. Answer: We have changed all tables accordingly.
- In Figure 1. which is very interesting and valuable , the association between hypodontia of maxillary PM2 and mandibulary PM2 is missed . If it is possible for authors it can be added. 3. Answer: We believe, that is it described as: " Similarly, the absence of PM2 in the mandible seemed to be independent of the absence of PM2 in the maxilla".
Reviewer 2 Report
By analyzing the panoramic radiograph, the authors studied the absence of second premolar (PM2) and third molar (M3) and the development stages of M3 in patients with tooth agenesis in orthodontic clinics and investigated the availability and best age period of M3 as donor teeth for auto-transplantation to replace congenitally absent PM2 in children and young adults. The study is clearly organized, and the manuscript is well written.
There are some points to be addressed:
1. In title and many places of manuscript: autotransplanation or auto-transplantation?
2. Page2 line88 “The exclusion criteria comprised the presence of hypodontia of more than 6 permanent teeth except M3.” If more than 6 teeth are missing, it should be termed oligodontia, here “hypodontia” should be changed to “tooth agenesis”.
3.Page2 line64 For the diagnostics of tooth agenesis, taking two panoramic radiographs in 1 month interval seems not necessary?
4. Add title for each table and put it on top of the table.
5. Page8 line240 Fig1 index seems wrong?
6. Page9 line 286 Where is Fig 2?
7. Please correct the ordinal numbers of Reference
Author Response
Dear Reviewer,
Many thanks for your time to review our paper and for your valuable comments to the manuscript.
The detailed responses are provided below:
1. In title and many places of manuscript: autotransplanation or auto-transplantation?
Answer: We have used the term autotransplanation as more common in the literature on tooth transplanation.
2. Page2 line88 “The exclusion criteria comprised the presence of hypodontia of more than 6 permanent teeth except M3.” If more than 6 teeth are missing, it should be termed oligodontia, here “hypodontia” should be changed to “tooth agenesis”.
Answer: We have included the suggested changes.
3.Page2 line64 For the diagnostics of tooth agenesis, taking two panoramic radiographs in 1 month interval seems not necessary?
Answer: We have used the same radiographs at 1-month interval only to check the reliability to diagnose congenital absence of PM2 and M3.
4. Add title for each table and put it on top of the table.
Answer: We haver included the requested changes.
5. Page8 line240 Fig1 index seems wrong?
Answer: It was a mistake and it is corrected now.
6. Page9 line 286 Where is Fig 2?
Answer: It was a mistake and it is corrected now.
7. Please correct the ordinal numbers of Reference
Answer: We have corrected the references accordingly.